# Chemical Composition, Larvicidal and Molluscicidal Activity of Essential Oils of Six Guava Cultivars Grown in Vietnam

**DOI:** 10.3390/plants12152888

**Published:** 2023-08-07

**Authors:** Huynh Van Long Luu, Huy Hung Nguyen, Prabodh Satyal, Van Hoa Vo, Gia Huy Ngo, Van The Pham, William N. Setzer

**Affiliations:** 1Institute of Applied Technology, Thu Dau Mot University, 06 Tran Van On, Thu Dau Mot City 820000, Vietnam; longlhv@tdmu.edu.vn; 2Center for Advanced Chemistry, Institute of Research and Development, Duy Tan University, 03 Quang Trung, Da Nang 550000, Vietnam; ngogiahuy@duytan.edu.vn; 3Department of Pharmacy, Duy Tan University, 03 Quang Trung, Da Nang 550000, Vietnam; hv697108@gmail.com; 4Aromatic Plant Research Center, 230 N 1200 E, Suite 100, Lehi, UT 84043, USA; psatyal@aromaticplant.org (P.S.); wsetzer@chemistry.uah.edu (W.N.S.); 5Laboratory of Ecology and Environmental Management, Science and Technology Advanced Institute, Van Lang University, Ho Chi Minh City 70000, Vietnam; phamvanthe@vlu.edu.vn; 6Faculty of Applied Technology, School of Technology, Van Lang University, Ho Chi Minh City 70000, Vietnam; 7Department of Chemistry, University of Alabama in Huntsville, Huntsville, AL 35899, USA

**Keywords:** *Aedes*, *Culex*, environmentally friendly, *Indoplanorbis exustus*, *Physa acuta*, *Psidium guajava*

## Abstract

Diseases transmitted by mosquitoes and snails cause a large burden of disease in less developed countries, especially those with low-income levels. An approach to control vectors and intermediate hosts based on readily available essential oils, which are friendly to the environment and human health, may be an effective solution for disease control. Guava is a fruit tree grown on a large scale in many countries in the tropics, an area heavily affected by tropical diseases transmitted by mosquitoes and snails. Previous studies have reported that the extracted essential oils of guava cultivars have high yields, possess different chemotypes, and exhibit toxicity to different insect species. Therefore, this study was carried out with the aim of studying the chemical composition and pesticide activities of six cultivars of guava grown on a large scale in Vietnam. The essential oils were extracted by hydrodistillation using a Clevenger-type apparatus for 6 h. The components of the essential oils were determined using gas-chromatography–mass-spectrometry (GC-MS) analysis. Test methods for pesticide activities were performed in accordance with WHO guidelines and modifications. Essential oil samples from Vietnam fell into two composition-based clusters, one of (*E*)-β-caryophyllene and the other of limonene/(*E*)-β-caryophyllene. The essential oils PG03 and PG05 show promise as environmentally friendly pesticides when used to control *Aedes* mosquito larvae with values of 24 h LC_50-*aegypti*_ of 0.96 and 0.40 µg/mL while 24 h LC_50-*albopictus*_ of 0.50 and 0.42 µg/mL. These two essential oils showed selective toxicity against *Aedes* mosquito larvae and were safe against the non-target organism *Anisops bouvieri*. Other essential oils may be considered as molluscicides against *Physa acuta* (48 h LC_50_ of 4.10 to 5.00 µg/mL) and *Indoplanorbis exustus* (48 h LC_50_ of 3.85 to 7.71 µg/mL) and with less toxicity to *A. bouvieri*.

## 1. Introduction

Guava (*Psidium guajava* L.) is a widely grown fruit tree around the world, especially in tropical and subtropical countries such as Pakistan, Mexico, Indonesia, Brazil, Bangladesh, Philippines [1,2], and Vietnam [3]. Parts such as leaves and roots of guava have been used in traditional medicine in many countries, such as Vietnam and China. Guava has been reported to have many beneficial pharmacological effects, such as diabetes, cardiovascular diseases, and cancer [4,5,6].

Mosquito-borne diseases are causing a large global burden [7], with an estimated half of the population of the world at risk [8,9,10], affecting more than 1 billion people and causing about 1 million deaths globally [11]. Mosquito control measures based on synthetic insecticides are gradually becoming less effective, as evidenced by the increasing global burden of disease [12,13] and the expansion of the distribution of mosquito species [7]. Moreover, chemical pesticides have shown many negative effects on human health, such as diabetes, reproductive disorders, neurological dysfunction, cancer, and respiratory disorders [14,15], and on the environment and ecosystems, such as pollution of water and soil sources and the risk of ecological imbalance and biodiversity [15]. Resistance to synthetic insecticides in mosquito species is a major concern that challenges mosquito-borne disease control programs [16,17,18,19].

Controlling viral vector diseases by releasing genetically modified mosquitoes is an approach that is receiving debate [20]. Modified mosquitoes have not yielded satisfactory results [21]; in addition, there are major disadvantages, such as high costs compared to benefits, and impacts on the ecosystem have not been evaluated [21,22,23,24].

*Indoplanorbis exustus* is an intermediate host for *Schistosoma indicum* species group, trematode parasites responsible for cattle schistosomiasis and human cercarial dermatitis [25]. This species has also been reported to be an intermediate host for the liver flukes, *Fasciola hepatica*, and *Fasciola gigantica* (Hyman, 1970), which cause immense harm to domestic animals in India [26]. Furthermore, it is an intermediate host for *Paramphistomum* species and *Echinostoma* species [27]. *Physella acuta* is an intermediate host of parasites that cause disease in humans, such as *Angiostrongylus cantonensis* (Chen, 1935) [28] and *Echinostoma revolutum* Looss (Echinostomida: Echinostomidae) [29]; and parasites that cause disease in animals, such as *Posthodiplostomum minimum* (Dubois, 1936) [30] and *Hypoderaeum conoideum* (Trematoda: Echinostomatidae). This species has become globally invasive [31,32].

Essential-oil-based pesticides are promising as biopesticides as an alternative to chemical pesticides with many advantages such as broad-spectrum efficacy, low toxicity to non-target organisms, and difficulty for the target organisms to develop resistance [33,34,35]. What is important when selecting solutions to control target species is sustainability and the correlation between cost and effectiveness. For essential oils, we believe that the source of raw materials from waste products of industrial plants will have high application prospects. The economic benefits from waste products can be a solution to the high-cost problem of essential-oil-based bioproducts.

There are several factors that can contribute to variations in essential oil composition. Genetic differences can result in phytochemical differences. For example, there are three general cultivars of *Cannabis sativa* L.: a high tetrahydrocannabinol (THC) cultivar, a high cannabidiol (CBD) cultivar, and a hybrid. In addition to cannabinoid concentration differences, differences in terpenoid concentrations have also been noted [36]. Geographical location can play an important role in the volatile phytochemistry of a species. There are obvious differences in tea tree (*Melaleuca alternifolia* Cheel) essential oil compositions based on geographical origin in Australia; an erpinen-4-ol-rich chemotype predominates in and around the Bungawalbin basin in the Casino area of northern New South Wales (NSW), the high 1,8-cineole chemotype predominates toward the southern end of the distribution around Grafton, NSW, and the terpinolene chemotype predominates in southern Queensland [37]. The phenological state and seasonality of the plant at the time of collection can have profound impacts on essential oil composition. Virginia mountain mint (*Pycnanthemum virginianum* Michx.) has demonstrated a significant seasonal variation in pulegone and isomenthone concentrations; pulegone concentrations diminish with concomitant increase in isomenthone concentration throughout the growing season [38]. Different methods of extraction can affect the volatile profiles of aromatic plants. Wide variations in compositions have been noted in nutmeg (*Myristica fragrans* Houtt.) depending on whether the volatile oil was obtained using steam distillation, high-vacuum distillation, head-space analysis, or supercritical fluid extraction [39]. With these potential variables in mind, the purpose of this research was to evaluate the yields, chemical compositions, and larvicidal and molluscicidal activities of six commercially grown guava cultivars in Vietnam. Furthermore, the toxicity of the essential oils to a non-target organism was also evaluated.

## 2. Results

### 2.1. Chemical Profiles of Essential Oils

The yield and main components (>4.0%) of essential oils of guava cultivars from Vietnam ranged from 0.4 to 0.53% (*v*/*w*) (Table 1); previous reports showed that essential oils of guava cultivars ranged from 0.11 to 0.9% (*v*/*w*) [40,41]. The full analytical results of six guava cultivars are available in the Appendix A.

### 2.2. Larvicidal Activities

The essential oils of the guava cultivars have been evaluated for larvicidal activity against *Ae. aegypti* (Table 2 and Table 3; major components summarized in Table 4), *Ae. albopictus* (Table 5 and Table 6; major components summarized in Table 7), and *Cx. fuscocephala* (Table 8). The essential oils pink flesh smooth skin guava (PG03) and Taiwan guava (PG05) were classified as “exceptionally active” against larvae of all three mosquito species with 24 h LC_50_ values < 10 μg/mL [42]. In addition, the essential oils pink flesh rough skin guava (PG04) and Queen guava (PG06) were shown to be “exceptionally active” against larvae of *Ae. aegypti* with 24 h LC_50_ values of 2.71 and 8.51 10 μg/mL, respectively [42]. Two essential oils, pink pearl guava (PG01) and white flesh guava (PG02), were shown to be “very active” against all three mosquito species [42].

### 2.3. Molluscicidal Activities

The molluscicidal activity of the six guava species did not follow the same trend as the larvicidal activity, i.e., the toxicity to each species of the essential oils was not significantly different, or the difference was not very large. The LC_90_ values of the essential oils at 48 h and 72 h were in the range of 6.65–10.54 and 5.04–8.12 μg/mL to *P. acuta* (Table 9, molluscicidal activities of major components summarized in Table 10); a range of 3.52–7.71 and 3.02–5.22 μg/mL to *I. exustus* (Table 11, molluscicidal activities of major components summarized in Table 12), respectively. There were no significant differences in molluscicidal activity between the essential oils. Based on the classification for the plant-based molluscicides, these essential oils were determined to be active (LC_90_ < 20 μg/mL) [42].

### 2.4. Toxicity of Essential Oils to the Non-Target Anisops Bouvieri

The essential oils exhibited similar trends in toxicity to *A. bouvieri* to the larvae of mosquito species (Table 13). The 90% lethal dose at 24 h of PG03 and PG05 essential oils for *Ae. aegypti*, *Ae. albopictus*, and *Cx. fuscocephala* were 1.75, 1.10, 6.40 and 0.68, 0.84, 6.12 μg/mL, respectively.

## 3. Discussion

### 3.1. Essential Oil Chemotypes

There are many different cultivars of *P. guajava,* and the volatile phytochemical profiles have shown wide variation. In order to place the six cultivars from Vietnam into a phytochemical context, a hierarchical cluster analysis (HCA) comparing the major components of 120 essential oils that were reported in the literature from 2015 to 2023, refs. [44,45,46,47,48,49,50,51,52,53,54,55,56,57,58,59,60,61,62,63,64,65,66,67,68,69,70,71,72,73,74,75,76,77] as well as the six specimens from Vietnam, was carried out (Figure 1). The cluster analysis revealed at least eight clusters. Cluster 1 is characterized by a high content of limonene and (*E*)-β-caryophyllene and contains two Vietnamese samples, PG02 and PG05. Cluster 2 is a group of (*E*)-β-caryophyllene, and essential oils PG01, PG03, PG04, and PG06 fall into this group. Cluster 3 is a group of components α-humulene, (*E*)-caryophyllene and is followed by selin-11-en-4α-ol. Cluster 4 is a group of components (*E*)-caryophyllene, α-selinene, and 14-hydroxy-9-*epi*-(*E*)-caryophyllene. Cluster 5 is a group of (*E*)-β-ocimene. Cluster 6 is a group of β-bisabolol and α-humulene. Cluster 7, made up of a single sample, is dominated by (*E*)-nerolidol. Finally, Cluster 8 is a group containing low concentrations of (*E*)-nerolidol.

### 3.2. Larvicidal Activities

The two cultivars, P03 (β-caryophyllene chemotype) and P05 (limonene/β-caryophyllene chemotype), were the most active essential oils in terms of larvicidal activity. The main components in the essential oils have been evaluated for larvicidal activity against larvae of two species of *Aedes* (Table 4 and Table 7). For the first time, the compounds globulol and nerolidol were evaluated for larvicidal activities, and globulol was stronger against both *Aedes* species than nerolidol. Significant differences in the larvicidal activity of compounds (*E*)-β-caryophyllene, α-pinene, and α-humulene have been reported by different research groups. The reasons for this difference may be due to reasons such as different health or developmental stages of the larvae or the protocol used by research groups [43]. β-Caryophyllene has shown less toxicity to *Ae. aegypti* (24 h LC_50_ of 111.66 μg/mL) than to *Ae. albopictus* (24 h LC_50_ of 30.11 μg/mL), and this trend is supported by Sobrinho et al. (2021) [78]. The (*R*)-(+)-limonene in this study exhibited larvicidal activity against two *Aedes* species in agreement with the majority of previous studies [79,80] and weaker than that reported by Dhinakaran et al. (24 h LC_50-*aegypti*_ of 11.88 μg/mL) [81]. A-Humulene in this study exhibited larvicidal activity against *Ae. Aegypti* (24 h LC_50_ of 48.19 μg/mL) and *Ae. Albopictus* (24 h LC_50_ of 31.49 μg/mL), which is consistent with previous studies [82,83,84]. Caryophyllene oxide in this study exhibited stronger larvicidal activities against the two *Aedes* species than previously reported results [80,85,86].

Mixtures of the main components in their respective proportions in the essential oils were evaluated for larvicidal activities against two species of *Aedes* (Table 4 and Table 7). All blends have shown much weaker toxicity than their respective essential oils. These results suggested that minor components were mainly responsible for the larvicidal activities of essential oils, which may have been through synergistic effects with the major components or between the minor components [87,88,89,90,91,92,93,94,95,96]. Some scientists believe that the main components within a certain concentration range will be mainly responsible for the biological activity of the essential oils; when the threshold concentration is exceeded, the effectiveness is attributed to the synergistic interaction of/with the minor compounds [97,98]. Scalerandi and co-authors found that insects preferentially oxidize the major terpenes in the mixture, while the minor terpenes act as toxicants [99]. Interestingly, the two essential oils, PG02 and PG05, were almost identical in terms of composition and content; however, PG05 has shown several times stronger toxicity to the larvae of three mosquito species than PG02. Van Vuuren and Viljoen have found that synergistic, antagonistic, or additive effects depend on the ratio and specific enantiomer [100]). Our previous study showed that the toxicity of a mixture of main components to the larvae of different mosquito species also varies [43]. Mendes and co-authors studied the chemical composition and larvicidal activity against *Ae. aegypti* of five guava cultivars from Brazil, three guava cultivars that fell into Cluster 2 showed weaker activity than the three cultivars (PG01, PG04, and PG06) from Vietnam [66]. The larvicidal activities of some minor compounds (<0.5% content) in the guava cultivars’ essential oils are presented in Table 14. However, none of these compounds alone can account for the strong larvicidal activities of the essential oils.

PG03 contained minor components, such as α-cadinol (LC_50-*albopictus*_ = 11.22 μg/mL) [108], α-bisabolol, *epi*-β-bisabolol (LC_50-*aegypti*_ = 15.83 μg/mL) [109], whereas a mixture of cadinol + α-bisabolol exhibited an LC_50-aegypti_ value of 2.53 μg/mL [110], suggesting that α-cadinol and α-bisabolol may be synergistic in larvicidal activities. All six essential oils contained α-copaene at concentrations above 2.0%, and it may be that these compounds that played an important role for the larvicidal activities via synergistic actions with the other components. *Hymenaea courbaril* fruit peel essential oil with the main components α-copaene (11.1%), spathulenol (10.1%) and β-selinene (8.2%) was effective against *Ae. aegypti* larvae with an LC_50_ value of 14.8 μg/mL [111], while also spathulenol and β-selinene both exhibited LC_50_ values > 100 μg/mL [104,107]. *Callicarpa sinuata* leaf essential oil contained two main components, α-copaene (12.6%) and α-humulene (24.8%), which have shown an LC_50-aegypti_ value of 25.86 μg/mL [112].

### 3.3. Molluscicidal Activities

All six *P. guajava* essential oils showed notable molluscicidal activities. Some of the major components in the essential oils have also shown strong molluscicidal activity against *P. acuta* (Table 10) and *I. exustus* (Table 12). Monoterpenes limonene and α-pinene have exhibited weaker toxicity than sesquiterpenoid compounds. The (*E*)*-*β-caryophyllene and maybe its synergistic actions with other constituents were responsible for the molluscicidal activity of the essential oils. Several previous studies have supported this trend. *Cannabis sativa* containing 18.7% of (*E*)-β-caryophyllene demonstrated greater toxicity to *P. acuta* than essential oils containing (*E*)-β-caryophyllene as a minor component [29,113]. (*E*)-Nerolidol did not exhibit toxicity against *Biomphalaria glabrata* at a concentration of 100 μg/mL [35]. Essential oils containing (*E*)-β-caryophyllene as the main constituent exhibited potent molluscicidal activities against *Gyraulus convexiusculus* and *Pomacea canaliculata* [42,43,114].

Several essential oils have been reported previously to exhibit molluscicidal activity against *P. acuta*, such as *Achillea millefolium* (48 h LC_50_ of 112.91 µg/mL), *Haplophyllum tuberculatum* (48 h LC_50_ of 73.70 µg/mL) [29], *C. sativa* (48 h LC_50_ of 35.37 μL/L), and *Humulus lupulus* (48 h LC_50_ of 118.65 μL/L) [113]. The compounds cypermethrin, permethrin, and fenvalerate were evaluated for molluscicidal activity for 24 to 96 h against *I. exustus*, LC_50_ values being 1.04 and 0.7, 1.55 and 0.94, and 1.7 and 0.07 µg/mL, respectively [115].

At the concentration of 1.75 μg/mL, the essential oils PG05 and PG03 were lethal to less than 1% of *A. bouvieri* at 48 h. Thus, the two essential oils are safe for *A. bouvieri* when used to control larvae of *Ae. aegypti* and *Ae. albopictus*. However, at concentrations of 6.12 and 6.4 μg/mL, nearly 90% of *A. bouvieri* were killed at 48 h. Similarly, other essential oils have shown non-selective toxicity to *Cx. fuscocephala* with SI values at 24 h between 0.7 and 1.0.

The two essential oils, PG03 and PG05, exhibited lower selective toxicities to the molluscs, *P. acuta* and *I. exustus*, with SI values at 48 h of 1.2, 0.9, and 1.74, 1.10, respectively. However, other essential oils showed less toxicity to *A. bouvieri* with SI values at 48 h of 2.7 to 4.6 and 3.15 to 4.51, respectively. Studies by Benelli et al. (2015) and Bedini et al. (2016) reported that the essential oils *A. millefolium*, *H. tuberculatum*, *C. sativa*, and *H. lupulus* exhibited non-selective toxicity to *Cloeon dipterum* when compared with the target species *Cx. pipiens, P. acuta*, and *Ae. albopictus* [29,113]. Benelli found that *Carlina acaulis* essential oil exhibited toxicity to *Daphnia magna* when compared with *Cx. quinquefasciatus* larvae [116]. Many previous studies have shown a tendency for essential oils to exhibit greater toxicity to *A. bouvieri* when compared with other non-target organisms such as *Diplonychus indicus*, *Gambusia affinis*, or *Poecilia reticulata* [116,117,118,119,120,121].

## 4. Materials and Methods

### 4.1. Plant Material

All six cultivars of guava (*Psidium guajava* L.) have been cultivated on a large scale in Cai Be district, Tien Giang province, Vietnam. Synthetic fertilizer NPK (synthetic N, P, and potassium; 16-16-8, *w*/*w*) was periodically applied in the months of January, April, June, and August of the year. Water has been irrigated by drip technology. Guava trees after four years of age and at two months after flowering (fruit-bearing time) were the subjects of this study. Mature leaves of six cultivars of guava were collected in October 2018 (Table 15). The collected leaves were transferred to laboratory conditions on the same day and were immediately used to extract the essential oils.

The plants were identified by Dr. Van The Pham. Six voucher specimens (from PG01 to PG06) have been deposited in the Herbarium of the Laboratory of the Institute of Applied Technology, Thu Dau Mot University, Vietnam.

According to the literature provided by the Southern Horticultural Research Institute (SOFRI) [3], Mitra and Irenaeus [122], this study contains six cultivars of guava, including ‘Nu hoang’ (Nữ hoàng) or ‘Queen guava’, ‘Ruot hong da san’ (Ruột hồng da sần), ‘Ruot hong da lang’ (Ruột hồng da láng), ‘Le Dai Loan’ (Lê Đài Loan) or Taiwan guava, ‘Se’ (Sẻ), and ‘Ruot trang’ (Ruột trắng). The ‘Nu hoang’ cultivar is characterized by white and soft flesh, few seeds, and orbicular fruits with a diameter of about 8 cm. The ‘Ruot hong da lang’ or ‘Pink flesh smooth skin’ and ‘Ruot hong da san’ or ‘Pink flesh rough skin’ are oval fruits with an average diameter of up to 9 cm, seedy, and crunchy flesh. The ‘Le Dai Loan’ cultivar is introduced from Taiwan, more or less orbicular fruits, soft and white flesh, and seedy. The ‘Se’ cultivar is a small orbicular fruit with an average diameter of about 4 cm, pink-red and thin flesh, and seedy. This cultivar is good for making juice. The ‘Ruot trang’ is characterized by short oval fruit, white and soft flesh, and seedy. Pictures of the fruit, flowers, and leaves of six guava cultivars are available in the Appendix A.

### 4.2. Hydrodistillation

The fresh leaves were chopped and hydrodistilled with a Clevenger apparatus (Witeg Labortechnik, Wertheim, Germany) for 6 h, 70 g of material and 500 mL of distilled water per trial, and the yield was calculated as the average of four consecutive trials. The essential oils (EOs) were dried over anhydrous Na_2_SO_4_, contained in brown 5 mL-vials, and stored at 4 °C until use.

### 4.3. Gas Chromatographic—Mass Spectral Analysis

Each of the EOs was analyzed by GC-MS using a Shimadzu GCMS-QP2010 Ultra (Shimadzu Scientific Instruments, Columbia, MD, USA) operated in the electron impact (EI) mode (electron energy = 70 eV), scan range = 40–400 atomic mass units, scan rate = 3.0 scans/s, and GC-MS solution software. The GC column was a ZB-5ms fused silica capillary column (Phenomenex, Torrance, CA, USA) (60 m length × 0.25 mm internal diameter) with a (5% phenyl)-polymethylsiloxane stationary phase and a film thickness of 0.25 μm. The carrier gas was helium with a column head pressure of 208 kPa and a flow rate of 2.00 mL/min. The injector temperature was 260 °C, and the ion source temperature was 260 °C. The GC oven temperature program was programmed for 50 °C initial temperature; the temperature increased at a rate of 2 °C/min to 260 °C and then held at 260 °C for 5 min. A 5% *w*/*v* solution of the sample in CH_2_Cl_2_ was prepared, and 0.1 μL was injected with a splitting mode (24.5:1).

Identification of the oil components was based on their retention indices determined by reference to a homologous series of *n*-alkanes (C_8_–C_40_) and by comparison of their mass spectral fragmentation patterns with those reported in the databases [123,124,125,126]. The percentages of each component in the EOs are reported as raw percentages based on total ion current without standardization.

### 4.4. Larvicidal Biassays

*Aedes aegypti* and *Ae. albopictus* have been continuously maintained in the laboratory of Duy Tan University. The adults were fed on 10% sucrose solution and blood-fed from white mice. Egg rafts of *Culex fuscocephala* were collected from rice fields in Hoa Vang, Da Nang (16°00′49″ N, 108°06′12″ E). Each egg raft was hatched separately in plastic trays containing tap water overnight; the 3rd instar and 4th early instar larvae were used for classification based on morphological characteristics [127,128]. The larvae that survived the trial were reared to adulthood and reclassified to confirm the initial identification. All larvae were fed on a mixture of dog food and yeast at a ratio of 3:1 (*w*/*w*). All developmental stages of mosquitos were maintained at 25 ± 2 °C, 65–75% relative humidity, and a 12:12 h light/dark cycle. The 3rd instar and 4th early instar larvae were used to evaluate the larvicidal activities of essential oils and purified compounds.

Tests for larvicidal activity were performed according to two protocols as described below. All tests were performed under laboratory conditions at 25 ± 2 °C, 65–75% relative humidity, and 12:12 h light/dark cycle. Dimethyl sulfoxide (DMSO, Sigma-Aldrich, Saint Louis, MO, USA) was used to prepare 1% (*w*/*v*) stock solutions and was also used as a negative control, and pyrethrin (Sigma-Aldrich) was used as a positive control. In both protocols, the solution level in the test beakers was always within the range of 5 to 10 cm [129].

Protocol 1: Tests for larvicidal activity were performed according to WHO guidelines [129] with modifications according to previous publications [42,43,83,114]. Twenty-five larvae were transferred into 250 mL beakers containing 150 mL of essential oil solutions at concentrations of 100, 50, 25, 12.5, 6.25, 3.125, 1.5, 0.75, 0.375, 0.2, and 0.1 μg/mL; each concentration was repeated four times. Larval mortality was determined at 24 h and 48 h of exposure.

Protocol 2: This protocol was performed in the same way as protocol 1, but the test solution volume was 50 mL in 150 mL beakers instead of 150 mL in 250 mL beakers, used for essential oils, major components, and major component mixtures. Testing of each pure compound and each major component mixture was performed twice on two different days, each time using a freshly prepared stock solution. After the conclusion of the two trials, the one with the stronger toxicity result was used to analyze the results.

### 4.5. Molluscicidal Bioassays

The adult snails of *P. acuta* (approximately 10 mm in length) were collected from aquarium cement tanks in Da Nang. The snails were acclimatized to laboratory conditions (25 ± 2 °C; 70 ± 5% RH, natural photoperiod) in a glass tank (50 cm wide, 100 cm long, 30 cm water level) for 48 h before testing and fed on the leaves of *Lactuca sativa* L.

Five adults were randomly selected and transferred to 200 mL beakers filled with 195 mL of double distilled water. Snails were exposed to essential oils at concentrations of 50, 25, 12.5, 6.25, and 3.125 μg/mL, replicated four times each. To prevent the snails from escaping, plastic Petri dishes were used to cover the top of the test beakers. Snails were allowed to recover after 24 h of exposure by transferring them to beakers containing only 195 mL of double-distilled water and identifying dead snails after the next 24 and 48 h. Snails were considered dead when there was no sign of a contraction response when probed with a needle [29,43,113]. Copper sulfate (Xilong Chemicals, Shantou, China) was used as a positive control in this experiment [43].

### 4.6. Toxicity on Anisops Bouvieri

Adults of *A. bouvieri* (Notonectidae) were collected from the wild in Da Nang city (16°00′22″ N; 108°15′45″ E) with a soft mesh and were maintained in glass tanks (60 cm long × 50 cm wide) at laboratory conditions (25 ± 2 °C, 65–75% relative humidity, 12:12 h light/dark cycle) for 48 h for familiarization before testing. The adults of *A. bouvieri* were identified based on morphology as described by Nieser [130], Ehamalar and Chandra [131]. Evaluation of the toxicity of essential oils against *A. bouvieri* was performed similar to protocol 1, using 20 adults for each repetition; concentrations of 100, 50, 25, 12.5, 6.25, and 3.125 μg/mL were used. The tests were performed under laboratory conditions at 25 ± 2 °C, 65–75% relative humidity, 12:12 h light/dark cycle. The Selectivity Index (SI) between the target organism and the non-target organism was calculated using the formula [132]:$$SI=\frac{24\;h\;{LC}_{50}\;of\;Anisops\;bouvieri}{24\;h\;{LC}_{50}\;of\;target\;organism}$$

### 4.7. Statistical Analysis

Lethality data were subjected to log-probit analysis [133] to obtain LC_50_ values, LC_90_ values, and 95% confidence limits using Minitab^®^ version 19.2020.1 (Minitab, LLC, State College, PA, USA). ANOVA testing was performed using Minitab^®^ version 19.2020.1 (Minitab, LLC, State College, PA, USA).

## 5. Conclusions

*Psidium guajava* is well known for its use as a source of fruit as well as medicinal benefits. In this work, the potential mosquito larvicidal activities and molluscicidal activities of essential oils from six cultivars of *P. guajava* growing in Vietnam have been explored. Two cultivars belong to a limonene/β-caryophyllene chemotype, while the other four were of a β-caryophyllene-rich chemotype. Two essential oils showed remarkable larvicidal activity, while all six essential oils were actively molluscicidal. The biological activities of the major components do not explain the activities of the essential oils, and the synergistic effects of minor components are likely responsible. Unfortunately, there are not enough data yet to parse out these effects. Additional research is needed to examine other chemotypes of *P. guajava*, seasonal and geographical variations in essential oil composition, and additional screening of essential oil components. It may be that with additional compositional data along with bioactivity data, a machine-learning approach may provide some insight into the synergistic effects of the essential oil components. Nevertheless, this study has demonstrated the larvicidal and molluscicidal potential of renewable *P. guajava* leaf essential oils.

## Figures and Tables

**Figure 1 plants-12-02888-f001:**
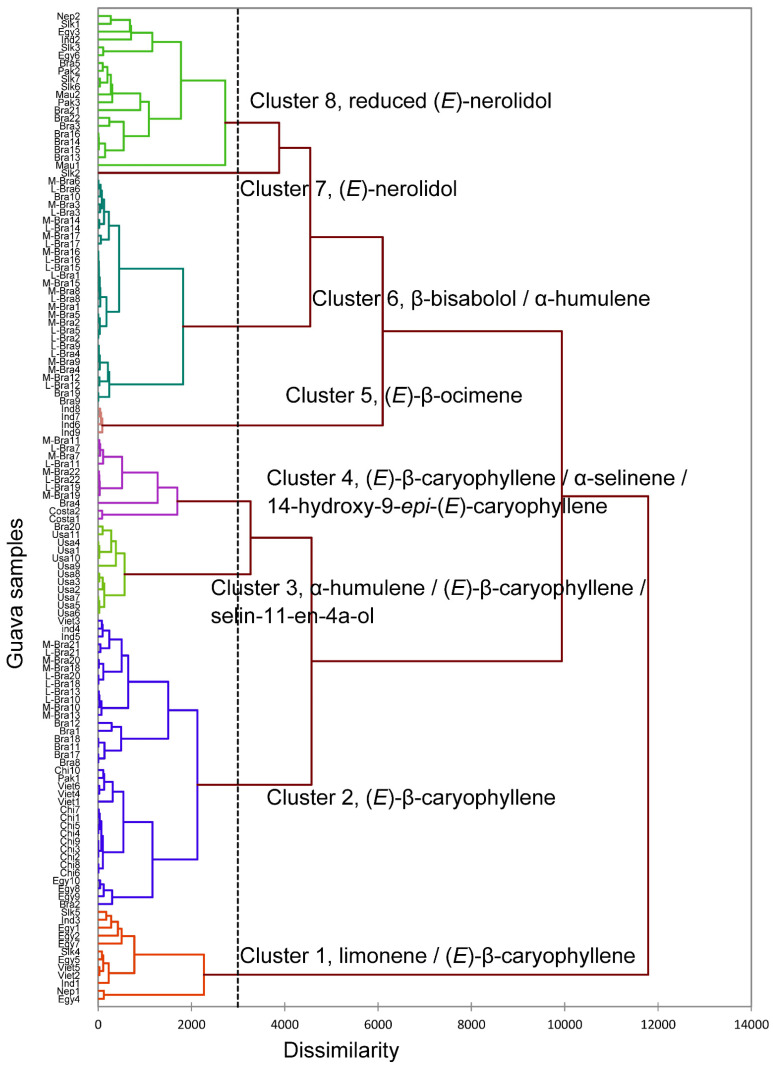
Dendrogram obtained from the agglomerative hierarchical cluster analysis of *Psidium guajava* essential oil compositions. Nep2 (Nepal) [44], Slk1 (Sri Lanka) [45], Egy3 (Egypt) [46], Ind2 (India) [47], Slk3 (Sri Lanka) [45], Egy6 (Egypt) [48], Bra5 (Brazil) [49], Pak2 (Pakistan) [50], Slk7 (Sri Lanka) [45], Slk6 (Sri Lanka) [45], Mau2 (Mauritius) [51], Pak3 (Pakistan) [50], Bra21 (Brazil) [52], Bra22 (Brazil) [52], Bra3 (Brazil) [53], Bra16 (Brazil) [54], Bra14 (Brazil) [55], Bra15 (Brazil) [55], Bra13 (Brazil) [55], Mau1 (Mauritius) [56], Slk2 (Sri Lanka) [45], M-Bra1→22 (Brazil) [57,58], L-Bra1→22 (Brazil) [57,58], Ind8 (India) [41], Ind7 (India) [41], Ind6 (India) [41], Ind9 (India) [41], Bra4 (Brazil) [59], Costa2 (Costa Rica) [60], Costa1 (Costa Rica) [60], Bra20 (Brazil) [52], USa11 (United States) [61], USa4 (United States) [62], USa1 (United States) [63], USa10 (United States) [61], USa9 (United States) [64], USa8 (United States) [62], USa3 (United States) [63], USa2 (United States) [63], USa7 (United States) [62], USa5 (United States) [62], USa6 (United States) [62], Viet3 (This study, PG03), Ind4 (India) [40], Ind5 (India) [65], Bra12 (Brazil) [66], Bra1 (Brazil) [67], Bra18 (Brazil) [54], Bra11 (Brazil) [66], Bra17 (Brazil) [54], Bra8 (Brazil) [66], Chi10 (China) [68], Pak1 (Pakistan) [69], Viet6 (This study, PG06), Viet4 (This study, PG04), Viet1 (This study, PG01), Chi1→9 (China) [70], Egy10 (Egypt) [48], Egy8 (Egypt) [48], Egy9 (Egypt) [48], Bra2 (Brazil) [71], Slk5 (Sri Lanka) [45], Ind3 (India) [72], Egy1 (Egypt) [73], Egy2 (Egypt) [74], Egy7 (Egypt) [48], Slk4 (Sri Lanka) [45], Egy5 (Egypt) [48], Viet5 (This study, PG05), Viet2 (This study, PG02), Ind1 (India) [75], Nep1 (Nepal) [76], Egy4 (Egypt) [77].

**Table 1 plants-12-02888-t001:** Main components (percent composition) of guava cultivars’ essential oils grown in Vietnam.

RI_(calc)_	RI_(db)_	Compound	PG01	PG02	PG03	PG04	PG05	PG06
%Yield (*v*/*w*)			0.51	0.43	0.40	0.48	0.43	0.53
933	933	α-Pinene	13.0	0.5	0.4	tr	0.3	0.1
1029	1030	Limonene	0.7	26.2	1.3	0.4	20.8	0.5
1375	1375	α-Copaene	2.4	4.1	2.7	2.4	4.2	5.3
1420	1417	(*E*)-β-Caryophyllene	13.9	20.4	21.7	30.0	24.8	27.8
1439	1439	Aromadendrene	7.5	2.9	3.0	5.9	3.0	3.5
1455	1454	α-Humulene	2.7	3.0	4.0	4.3	3.6	4.7
1560	1560	(*E*)-Nerolidol	1.4	0.1	13.7	8.6	tr	7.8
1583	1587	Caryophyllene oxide	8.1	3.7	11.4	5.7	2.4	5.3
1587	1590	Globulol	11.8	5.5	6.4	10.9	5.9	6.0

**Table 2 plants-12-02888-t002:** Larvicidal activity of guava cultivars’ essential oils against *Aedes aegypti* (μg/mL) (Protocol 1).

Material	LC_50_ (95% Limits)	LC_90_ (95% Limits)	χ^2^	*p*	SI
		24 h			
PG01	17.53 (15.96–19.26)	30.17 (26.61–35.83)	4.6216	0.202	1.3
PG02	16.79 (15.23–18.51)	30.39 (26.60–36.47)	2.4262	0.489	0.9
PG03	0.96 (0.87–1.06)	1.75 (1.53–2.10)	8.1947	0.316	7.0
PG04	2.71 (2.48–2.91)	3.90 (3.56–4.48)	15.4765	0.009	5.7
PG05	0.40 (0.36–0.43)	0.68 (0.60–0.81)	4.2866	0.746	14.9
PG06	8.51 (7.81–9.37)	10.71 (9.69–12.38)	0.0085	1.000	1.9
		48 h			
PG01	15.39 (13.93–17.02)	28.92 (25.17–34.94)	5.4934	0.139	1.4
PG02	14.75 (13.26–16.42)	30.35 (26.08–37.24)	4.3559	0.226	1.0
PG03	0.68 (0.63–0.72)	0.91 (0.84–1.05)	0.2337	1.000	7.5
PG04	2.39 (2.12–2.60)	3.02 (2.78–3.33)	0.0113	1.000	5.4
PG05	0.36 (0.32–0.39)	0.65 (0.57–0.79)	4.4003	0.733	12.3
PG06	7.61 (7.10–8.45)	9.74 (8.71–12.01)	0.0338	0.998	1.9

**Table 3 plants-12-02888-t003:** Larvicidal activity of guava cultivars’ essential oils against *Aedes aegypti* (μg/mL) (Protocol 2).

Material	LC_50_ (95% Limits)	LC_90_ (95% Limits)	χ^2^	*p*	SI
		24 h			
PG01	24.87 (23.55–26.24)	33.16 (30.36–39.61)	0.97545	0.987	0.93
PG02	24.18 (22.72–25.62)	33.59 (30.79–38.93)	1.4240	0.964	0.66
PG03	1.83 (1.71–1.98)	2.67 (2.41–3.11)	0.8445	0.991	3.67
PG04	17.68 (16.42–19.03)	24.07 (22.10–26.90)	0.1317	1.000	0.87
PG05	0.97 (0.90–1.04)	1.36 (1.24–1.56)	0.3309	0.999	6.15
PG06	16.35 (15.25–17.72)	21.35 (19.44–24.42)	0.0489	1.000	0.98
		48 h			
PG01	24.25 (22.67–25.90)	35.73 (32.49–41.29)	3.4661	0.748	0.89
PG02	23.71 (22.19–25.22)	33.92 (31.04–38.98)	1.8364	0.934	0.63
PG03	1.49 (1.37–1.62)	2.56 (2.28–2.99)	1.9566	0.924	3.42
PG04	17.33 (16.12–18.68)	23.58 (21.62–26.67)	0.1299	1.000	0.75
PG05	0.84 (0.77–0.91)	1.47 (1.31–1.73)	6.5423	0.365	5.25
PG06	15.42 (14.45–16.67)	20.82 (18.87–24.14)	0.1705	1.000	0.94

**Table 4 plants-12-02888-t004:** Larvicidal activity of major compounds of guava cultivars’ essential oils against *Aedes aegypti* (μg/mL) (Protocol 2).

Compounds	LC_50_ (95% Limits)	LC_90_ (95% Limits)	χ^2^	*p*
	24 h			
Caryophyllene oxide	39.65 (35.83–42.53)	49.41 (46.31–53.36)	0.011	1.00
α-Humulene	48.19 (44.33–52.29)	87.64 (78.81–100.02)	1.890	0.596
(*E*)-β-Caryophyllene	111.66 (105.55–118.0)	160.10 (151.39–170.85)	3.782	0.436
α-Pinene	12.94 (11.77–14.23)	26.48 (23.13–31.61)	3.0799	0.379
Globulol	11.13 (10.28–11.74)	14.53 (13.62–16.27)	0.1566	0.984
(*E*)-Nerolidol	36.22 (33.03–39.79)	75.25 (65.81–89.34)	9.1304	0.058
Limonene	17.66 (16.45–18.97)	23.62 (22.03–25.73)	0.784	0.941
(*E*)-β-Caryophyllene (14.26%), α-pinene (13.28%), globulol (11.98%), caryophyllene oxide (8.34%), α-humulene (2.27%). (PG01)	14.79 (13.79–15.95)	22.28 (20.11–25.76)	1.7426	0.883
Limonene (26.5%), (*E*)-β-caryophyllene (20.59%), globulol (5.24%), caryophyllene oxide (3.27%), α-humulene (3.1%). (PG02)	37.59 (34.75–40.31)	49.53 (46.04–54.12)	0.4941	0.992
(*E*)-β-Caryophyllene (22.09%), (*E*)-nerolidol (13.97%), caryophyllene oxide (11.51%), globulol (6.36%), α-humulene (4.0%). (PG03)	56.18 (53.24–60.83)	73.41 (66.18–90.04)	0.1527	1.000
(*E*)-β-Caryophyllene (30.16%), globulol (10.97%), (*E*)-nerolidol (8.72%), caryophyllene oxide (5.76%), α-humulene (4.24%). (PG04).	60.05 (56.44–65.99)	76.49 (68.83–92.86)	0.0367	1.000
(*E*)*-*β-Caryophyllene (25.32%), limonene (21.23%), globulol (6.05%), α-humulene (3.62%), caryophyllene oxide (2.45%). (PG05)	50.00 (47.11–53.07)	69.60 (63.50–81.49)	2.2110	0.819
(*E*)*-*β-Caryophyllene (28.06%), globulol (6.07%), caryophyllene oxide (5.34%), α-humulene (4.76%). (PG06)	78.62 (70.97–84.46)	97.61 (91.03–105.55)	0.0089	1.000
	48 h			
Caryophyllene oxide	37.92 (34.73–40.82)	47.94 (44.58–52.34)	0.015	1.000
α-Humulene	36.22 (33.15–39.51)	70.58 (62.82–81.67)	5.124	0.163
(*E*)-β-Caryophyllene	94.43 (88.37–100.84)	145.91 (136.85–157.04)	1.821	0.769
α-Pinene	11.56 (10.39–12.86)	28.21 (24.10–34.62)	7.4501	0.059
Globulol	10.20 (9.44–10.88)	13.82 (12.87–15.21)	0.1930	0.979
(*E*)-Nerolidol	33.19 (30.17–36.57)	72.64 (63.20–86.67)	8.9965	0.061
Limonene	17.43 (16.24–18.74)	23.17 (21.58–25.28)	0.664	0.956
(*E*)*-*β-Caryophyllene (14.26%), α-pinene (13.28%), globulol (11.98%), caryophyllene oxide (8.34%), α-humulene (2.27%). (PG01)	12.25 (11.32–13.25)	20.46 (18.35–23.72)	1.4036	0.924
Limonene (26.5%), (*E*)*-*β-caryophyllene (20.59%), globulol (5.24%), caryophyllene oxide (3.27%), α-humulene (3.1%). (PG02)	32.63 (30.43–35.10)	45.92 (41.96–51.81)	0.8873	0.971
(*E*)*-*β-Caryophyllene (22.09%), (*E*)-nerolidol (13.97%), caryophyllene oxide (11.51%), globulol (6.36%), α-humulene 4.0%). (PG03)	52.43 (49.28–56.02)	75.11 (68.04–88.08)	1.3948	0.925
(*E*)*-*β-Caryophyllene (30.16%), globulol (10.97%), (*E*)-nerolidol (8.72%), caryophyllene oxide (5.76%), α-humulene (4.24%). (PG04)	56.54 (53.00–60.78)	81.69 (73.70–95.68)	9.1397	0.104
(*E*)*-*β-Caryophyllene (25.32%), limonene (21.23%), globulol (6.05%), α-humulene (3.62%), caryophyllene oxide (2.45%). (PG05)	45.74 (42.72–48.42)	62.88 (58.16–71.38)	0.6529	0.985
(*E*)-β-Caryophyllene (28.06%), globulol (6.07%), caryophyllene oxide (5.34%), α-humulene (4.76%). (PG06)	74.41 (68.57–80.05)	95.64 (88.64–105.11)	0.0072	1.000

**Table 5 plants-12-02888-t005:** Larvicidal activity of guava cultivars’ essential oils against *Aedes albopictus* (μg/mL) (Protocol 1).

Material	LC_50_ (95% Limits)	LC_90_ (95% Limits)	χ^2^	*p*	SI
		24 h			
PG01	30.99 (28.35–33.76)	60.43 (54.05–69.40)	11.0698	0.271	0.8
PG02	24.32 (20.04–26.80)	53.86 (46.98–64.07)	11.7142	0.164	0.7
PG03	0.50 (0.46–0.55)	1.05 (0.91–1.29)	3.2586	0.860	13.4
PG04	11.04 (10.23–11.87)	17.24 (15.64–19.73)	1.4640	0.984	1.4
PG05	0.42 (0.39–0.46)	0.82 (0.72–0.99)	2.3287	0.939	14.2
PG06	18.88 (17.20–20.71)	40.81 (35.91–47.79)	25.5436	0.001	0.8
		48 h			
PG01	25.55 (23.23–28.01)	54.23 (47.98–63.08)	12.2525	0.199	0.8
PG02	21.39 (19.71–23.18)	36.20 (32.47–41.86)	7.9069	0.095	0.7
PG03	0.42 (0.38–0.46)	0.87 (0.76–1.06)	2.0098	0.959	12.1
PG04	8.10 (7.49–8.67)	12.98 (11.67–14.98)	14.3258	0.046	1.6
PG05	0.36 (0.33–0.39)	0.69 (0.61–0.82)	0.9844	0.995	12.3
PG06	8.83 (8.07–9.74)	17.94 (15.76–21.20)	4.3245	0.827	1.6

**Table 6 plants-12-02888-t006:** Larvicidal activity of guava cultivars’ essential oils against *Aedes albopictus* (μg/mL) (Protocol 2).

Material	LC_50_ (95% Limits)	LC_90_ (95% Limits)	χ^2^	*p*	SI
		24 h			
PG01	23.53 (22.05–24.76)	31.19 (28.90–36.16)	0.4124	0.999	0.99
PG02	24.72 (23.28–26.22)	34.41 (31.44–40.16)	1.9117	0.928	0.64
PG03	1.59 (1.50–1.70)	2.23 (2.01–2.65)	0.1651	1.000	4.22
PG04	14.25 (13.06–15.56)	26.38 (23.31–31.07)	4.2855	0.232	1.08
PG05	1.42 (1.32–1.51)	2.03 (1.86–2.34)	1.4142	0.965	4.20
PG06	18.16 (16.90–19.57)	27.22 (25.22–29.83)	3.0006	0.392	0.88
		48 h			
PG01	21.99 (20.47–23.45)	31.65 (29.1135.73)	1.0574	0.983	0.96
PG02	23.34 (21.74–24.96)	34.98 (31.83–40.19)	3.0652	0.801	0.64
PG03	1.50 (1.41–1.61)	2.24 (2.02–2.61)	0.9116	0.989	3.4
PG04	9.62 (8.84–10.46)	16.71 (14.93–19.39)	7.0871	0.069	1.35
PG05	1.34 (1.24–1.44)	2.07 (1.87–2.37)	2.7200	0.843	3.29
PG06	16.80 (15.62–18.12)	25.25 (23.40–27.64)	6.6641	0.083	0.86

**Table 7 plants-12-02888-t007:** Larvicidal activity of major compounds of guava cultivars’ essential oils against *Aedes albopictus* (μg/mL) (Protocol 2).

Compounds	LC_50_ (95% Limits)	LC_90_ (95% Limits)	χ^2^	*p*
	24 h			
Caryophyllene oxide	38.68 (35.84–41.44)	53.28 (49.31–58.93)	0.212	0.976
α-Humulene	31.49 (28.62–34.67)	65.14 (56.73–78.08)	8.186	0.042
(*E*)-β-Caryophyllene	30.11 (27.65–32.81)	53.88 (47.80–63.20)	1.865	0.601
α-Pinene	23.05 (21.25–24.99)	39.37 (35.23–45.72)	1.1294	0.890
Globulol	17.40 (16.15–18.76)	25.57 (23.28–28.96)	0.6607	0.882
(*E*)-Nerolidol	21.45 (19.94–22.89)	30.57 (28.20–34.29)	0.7489	0.862
Limonene	12.92 (12.20–13.75)	17.96 (16.32–21.16)	3.4336	0.329
(*E*)-β-Caryophyllene (14.26%), α-pinene (13.28%), globulol (11.98%), caryophyllene oxide (8.34%), α-humulene (2.27%). (PG01)	23.66 (21.67–25.83)	43.98 (38.91–51.73)	6.4225	0.093
Limonene (26.5%), (*E*)-β-caryophyllene (20.59%), globulol (5.24%), caryophyllene oxide (3.27%), α-humulene (3.1%). (PG02)	56.18 (53.24–60.83)	73.41 (66.18–90.04)	6.9076	0.075
(*E*)-β-Caryophyllene (22.09%), (*E*)-nerolidol (13.97%), caryophyllene oxide (11.51%), globulol (6.36%), α-humulene 4.0%). (PG03)	66.38 (61.19–73.82)	80.74 (72.80–95.02)	0.0018	1.000
(*E*)-β-Caryophyllene (30.16%), globulol (10.97%), (*E*)-nerolidol (8.72%), caryophyllene oxide (5.76%), α-humulene (4.24%). (PG04)	70.71 (63.33–78.95)	83.46 (75.02–96.00)	0.0001	1.000
(*E*)-β-Caryophyllene (25.32%), limonene (21.23%), globulol (6.05%), α-humulene (3.62%), caryophyllene oxide (2.45%). (PG05)	67.39 (61.85–75.02)	81.39 (73.38–95.26)	0.0010	1.000
(*E*)*-*β-Caryophyllene (28.06%), globulol (6.07%), caryophyllene oxide (5.34%), α-humulene (4.76%). (PG06)	69.34 (63.76–75.89)	85.26 (77.72–96.64)	0.0020	1.000
	48 h			
Caryophyllene oxide	33.95 (31.55–36.61)	49.37 (44.93–56.02)	2.136	0.545
α-Humulene	26.44 (24.0–29.13)	55.80 (48.53–66.95)	5.662	0.129
β-Caryophyllene	25.70 (23.46–28.17)	50.26 (44.15–59.60)	3.258	0.354
α-Pinene	22.91 (21.13–24.80)	38.51 (34.53–44.62)	10.5692	0.032
Globulol	13.97 (13.25–15.10)	18.28 (16.49–22.39)	6.3228	0.097
(*E*)-Nerolidol	19.18 (17.73–20.71)	30.16 (27.32–34.41)	2.0822	0.556
Limonene	11.83 (11.04–12.62)	17.38 (15.85–20.00)	2.5806	0.461
(*E*)-β-Caryophyllene (14.26%), α-pinene (13.28%), globulol (11.98%), caryophyllene oxide (8.34%), α-humulene (2.27%). (PG01)	23.01 (21.04–25.16)	43.64 (38.52–51.44)	4.4156	0.220
Limonene (26.5%), (*E*)-β-caryophyllene (20.59%), globulol (5.24%), caryophyllene oxide (3.27%), α-humulene (3.1%). (PG02)	52.43 (49.28–56.02)	75.11 (68.04–88.08)	6.0500	0.109
(*E*)-β-Caryophyllene (22.09%), (*E*)-nerolidol (13.97%), caryophyllene oxide (11.51%), globulol (6.36%), α-humulene 4.0%). (PG03)	62.08 (58.04–68.59)	77.92 (70.14–93.73)	0.0154	0.999
(*E*)-β-Caryophyllene (30.16%), globulol (10.97%), (*E*)-nerolidol (8.72%), caryophyllene oxide (5.76%), α-humulene (4.24%). (PG04)	66.38 (61.19–73.82)	80.74 (72.80–95.02)	0.0018	1.000
(*E*)-β-Caryophyllene (25.32%), limonene (21.23%), globulol (6.05%), α-humulene (3.62%), caryophyllene oxide (2.45%). (PG05)	64.17 (59.63–71.17)	79.31 (71.46–94.42)	0.0057	1.000
(*E*)-β-Caryophyllene (28.06%), globulol (6.07%), caryophyllene oxide (5.34%), α-humulene (4.76%). (PG06).	67.40 (62.41–73.67)	84.20 (76.63–96.30)	0.0057	1.000

**Table 8 plants-12-02888-t008:** Larvicidal activity of guava cultivars’ essential oils against *Culex fuscocephala* (μg/mL).

Material	LC_50_ (95% Limits)	LC_90_ (95% Limits)	χ^2^	*p*	SI
		24 h			
PG01	27.61 (25.06–30.44)	58.75 (51.09–70.40)	7.8329	0.166	0.8
PG02	23.64 (21.60–25.81)	46.13 (41.07–53.34)	7.3769	0.598	1.0
PG03	4.27 (3.91–4.55)	6.37 (5.77–7.27)	0.9294	0.968	1.6
PG04	21.33 (19.23–23.59)	51.89 (45.25–61.37)	9.3878	0.402	0.7
PG05	4.38 (4.07–4.72)	6.11 (5.59–6.86)	0.2027	0.999	1.4
PG06	21.90 (19.58–24.42)	62.65 (53.78–75.48)	15.4051	0.080	0.7
Permethrin	0.0024 (0.0022–0.0026)	0.0037 (0.0034–0.0043)	2.1866	0.335	Nt
		48 h			
PG01	12.67 (11.56–13.89)	24.90 (21.91–29.38)	6.4116	0.268	1.7
PG02	7.97 (7.39–8.61)	12.33 (11.12–14.18)	1.8322	0.872	1.9
PG03	3.30 (3.05–4.57)	5.42 (4.85–6.32)	0.9017	0.970	1.5
PG04	11.08 (10.23–11.97)	18.17 (16.37–20.93)	12.2851	0.198	1.2
PG05	3.94 (3.68–4.25)	5.63 (5.11–6.47)	0.4118	0.995	1.1
PG06	10.43 (9.51–11.41)	20.19 (17.84–23.68)	17.7924	0.038	1.2
Permethrin	0.0023 (0.0022–0.0025)	0.0036 (0.0032–0.0041)	2.1010	0.350	Nt

Nt: Not tested.

**Table 9 plants-12-02888-t009:** Molluscicidal activity of guava cultivars’ essential oils against *Physa acuta*.

Essential Oil	LC_50_ (95% Limits)	LC_90_ (95% Limits)	χ2	*p*	SI
		48 h			
PG01	4.56 (3.65–5.69)	9.17 (7.07–14.74)	1.9734	0.741	4.6
PG02	5.63 (4.55–6.95)	10.54 (8.26–16.75)	5.5028	0.239	2.7
PG03	4.10 (3.39–4.99)	6.65 (5.38–10.14)	4.1057	0.392	1.2
PG04	4.89 (3.89–6.14)	10.24 (7.82–16.70)	0.6372	0.959	2.7
PG05	4.72 (3.77–5.91)	9.70 (7.44–15.71)	0.5504	0.968	0.9
PG06	5.00 (4.08–5.85)	7.05 (6.00–9.78)	0.0768	0.999	2.9
		72 h			
PG01	3.54 (3.00–4.30)	5.27 (4.33–8.75)	5.5430	0.236	Nd
PG02	4.25 (3.43–5.27)	8.12 (6.34–12.88)	5.6605	0.226	Nd
PG03	3.33 (2.74–4.04)	5.54 (4.46–8.81)	5.9707	0.201	Nd
PG04	4.12 (3.33–5.07)	7.56 (5.95–11.90)	0.9965	0.910	Nd
PG05	3.32 (2.79–4.00)	5.04 (4.14–8.39)	2.2865	0.587	Nd
PG06	3.66 (3.11–4.46)	6.37 (4.42–8.81)	8.1325	0.087	Nd

Nd: not defined.

**Table 10 plants-12-02888-t010:** Molluscicidal activity of main compounds of guava cultivars’ essential oils against *Physa acuta*.

Compound	LC_50_ (95% Limits)	LC_90_ (95% Limits)	χ^2^	*p*
		48 h		
Caryophyllene oxide	5.78 (4.86–6.92)	8.96 (7.38–13.42)	0.50	0.921 [43]
α-Humulene	7.24 (6.00–8.67)	11.88 (9.71–17.50)	0.62	0.887 [43]
(*E*)-β-Caryophyllene	9.58 (7.79–11.72)	18.08 (14.32–27.14)	0.88	0.829 [43]
Limonene	14.17 (12.08–17.23)	20.88 (17.18–34.70)	4.84	0.184
α-Pinene	18.98 (15.48–23.21)	32.80 (26.18–51.34)	1.20	0.549
CuSO_4_ (positive control)	0.66 (0.55–0.80)	0.85 (0.72–1.17)	0.00	0.998
		72 h		
Caryophyllene oxide	4.04 (3.43–4.96)	5.58 (4.64–8.65)	0.01	1.000
α-Humulene	5.68 (4.67–6.69)	8.49 (7.10–13.40)	0.44	0.931
(*E*)-β-Caryophyllene	7.93 (6.55–9.67)	13.09 (10.54–20.32)	0.81	0.846
Limonene	10.58 (8.70–12.66)	16.81 (13.81–25.28)	0.71	0.871
α-Pinene	18.32 (14.86–22.56)	32.90 (25.95–52.62)	1.67	0.433
CuSO_4_ (positive control)	0.58 (0.50–0.72)	0.82 (0.68–1.29)	0.07	0.968

**Table 11 plants-12-02888-t011:** Molluscicidal activity of guava cultivars’ essential oils against *Indoplanorbis exustus* (µg/mL).

Essential Oil	LC_50_ (95% Limits)	LC_90_ (95% Limits)	χ^2^	*p*	SI
		48 h			
PG01	7.71 (6.11–9.75)	16.50 (12.43–28.10)	1.4873	0.685	3.01
PG02	3.52 (2.71–4.47)	8.0 (5.98–13.91)	1.5924	0.661	4.51
PG03	3.85 (2.85–5.05)	10.92 (7.74–20.94)	3.8332	0.280	1.74
PG04	4.77 (3.93–5.68)	7.12 (5.95–10.24)	0.1875	0.980	3.24
PG05	5.41 (4.18–6.95)	13.23 (9.68–23.55)	3.3713	0.338	1.10
PG06	5.07 (4.07–6.28)	9.70 (7.56–15.58)	3.7764	0.287	3.15
CuSO_4_ (positive control)	0.28 (0.23–0.33)	0.43 (0.35–0.64)	0.3618	0.948	Nd
		72 h			
PG01	5.22 (4.03–6.69)	12.63 (9.28–22.36)	2.5880	0.460	Nd
PG02	3.02 (2.27–3.86)	7.19 (5.32–12.91)	1.3858	0.709	Nd
PG03	3.02 (2.27–3.86)	7.17 (5.31–12.85)	4.4832	0.214	Nd
PG04	4.24 (3.56–5.12)	6.18 (5.12–9.04)	0.0981	0.992	Nd
PG05	3.69 (2.97–4.55)	6.81 (5.34–11.02)	3.4468	0.328	Nd
PG06	3.96 (3.22–4.87)	7.03 (5.56–11.15)	1.5521	0.670	Nd
CuSO_4_ (positive control)	0.27 (0.22–0.32)	0.43 (0.35–0.65)	0.5844	0.900	Nd

Nd: not defined.

**Table 12 plants-12-02888-t012:** Molluscicidal activity of main compounds of guava cultivars’ essential oils against *Indoplanorbis exustus*.

Compound	LC_50_ (95% Limits)	LC_90_ (95% Limits)	χ^2^	*p*
		48 h		
Caryophyllene oxide	12.50 (10.21–15.29)	22.12 (17.57–34.93)	0.8787	0.928
α-Humulene	12.50 (10.21–15.29)	22.12 (17.57–34.93)	0.8782	0.928
(*E*)-β-Caryophyllene	13.38 (10.96–16.38)	23.54 (18.70–37.21)	0.3162	0.989
Limonene	22.56 (18.28–27.80)	41.94 (32.94–66.34)	0.8084	0.937
α-Pinene	16.48 (12.74–21.28)	42.53 (30.90–74.26)	6.5045	0.165
CuSO_4_ (positive control)	0.28 (0.23–0.33)	0.43 (0.35–0.64)	0.3618	0.948
		72 h		
Caryophyllene oxide	9.47 (7.66–11.65)	17.34 (13.68–27.21)	0.8169	0.936
α-Humulene	11.67 (9.52–14.25)	20.55 (16.37–32.25)	3.6023	0.463
(*E*)-β-Caryophyllene	10.94 (9.0–13.12)	17.61 (14.41–26.84)	1.1960	0.879
Limonene	21.06 (17.10–25.85)	38.28 (30.26–59.81)	0.4518	0.978
α-Pinene	10.45 (8.12–13.32)	24.88 (18.46–42.40)	2.8622	0.581
CuSO_4_ (positive control)	0.27 (0.22–0.32)	0.43 (0.35–0.65)	0.5844	0.900

**Table 13 plants-12-02888-t013:** Toxicity of Guava cultivars’ essential oils against *Anisops bouvieri* (μg/mL).

Material	LC_50_ (95% Limits)	LC_90_ (95% Limits)	χ2	*p*
		24 h		
PG01	23.25 (21.22–25.39)	39.50 (35.21–46.11)	14.0122	0.122
PG02	15.87 (14.49–17.40)	26.88 (23.80–31.70)	2.2395	0.987
PG03	6.71 (6.21–7.26)	10.05 (8.98–11.98)	10.5384	0.309
PG04	15.46 (14.00–17.07)	28.92 (25.36–34.40)	10.6390	0.301
PG05	5.97 (5.56–6.34)	8.20 (7.35–9.61)	0.5427	1.000
PG06	15.97 (14.66–17.44)	25.20 (22.47–29.54)	0.5132	1.000
		F		
PG01	21.05 (19.06–23.20)	39.68 (34.96–46.78)	17.4342	0.042
PG02	14.94 (13.59–16.43)	26.49 (23.34–31.43)	3.1997	0.956
PG03	5.10 (4.68–5.52)	7.48 (6.80–8.56)	0.6034	1.000
PG04	13.02 (11.76–14.40)	25.29 (22.06–30.29)	12.9108	0.167
PG05	4.41 (4.06–4.80)	6.22 (5.64–7.11)	0.1174	1.000
PG06	14.50 (13.35–15.80)	22.70 (201.21–26.84)	0.7521	1.000

**Table 14 plants-12-02888-t014:** Summary of the larvicidal activities of compounds with concentrations less than 0.5% (or isomer) in guava cultivars’ essential oils *.

Compound	24 h LC_50_ (μg/mL)	Mosquito	Ref.
Myrcene	27.9	*Aedes aegypti*	[80]
	35.8	*Aedes aegypti*	[101]
	39.51	*Aedes aegypti*	[102]
	23.5	*Aedes albopictus*	[80]
	27.0	*Aedes albopictus*	[101]
	35.98	*Aedes albopictus*	[102]
	41.31	*Culex pipiens pallens*	[102]
1,8-Cineole	73.30	*Aedes aegypti*	[102]
	73.50	*Aedes albopictus*	[102]
	72.88	*Culex pipiens pallens*	[102]
(*Z*)-β-ocimene	28.35	*Aedes aegypti*	[103]
	33.72	*Aedes albopictus*	[103]
	31.52	*Culex quinquefasciatus*	[103]
	37.13	*Culex tritaeniorhynchus*	[103]
Aromadendrene	>150	*Aedes aegypti*	[102]
	129.21	*Aedes albopictus*	[102]
β-Selinene	136.06	*Aedes aegypti*	[104]
	151.74	*Aedes albopictus*	[104]
(*Z*)-γ-Bisabolene	2.26	*Aedes aegypti*	[105]
	4.50	*Aedes albopictus*	[105]
	2.47	*Culex quinquefasciatus*	[105]
	4.87	*Culex tritaeniorhynchus*	[105]
δ-Cadinene	17.91	*Aedes aegypti*	[106]
	19.50	*Culex quinquefasciatus*	[106]
Spathulenol	>100	*Aedes aegypti*	[107]
α-Cadinol	11.22	*Aedes albopictus*	[108]
	12.28	*Culex tritaeniorhynchus*	[108]
*epi*-β-bisabolol	15.83	*Aedes aegypti*	[109]
	17.27	*Culex quinquefasciatus*	[109]
τ-muurolol + α-cadinol + α-bisabolol (16:21:46, %/%)	2.98	*Aedes aegypti*	[110]
τ-muurolol + α-cadinol + α-bisabolol (0:31:54, %/%)	2.53	*Aedes aegypti*	[110]

* For comparison purposes, only the lowest LC_50_ values have been selected.

**Table 15 plants-12-02888-t015:** Information about six Vietnamese cultivars of *Psidium guajava* L.

Vietnamese Cultivars ^1,2^	Vietnamese Name	English Name	Collection Site (Cai Be District, Tien Giang Province)	Voucher Number
Se	Ổi sẻ đỏ	Pink Pearl Guava	(10°24′44″ N, 105°52′4″ E, morn. 10 m)	PG01
Ruot trang	Ổi trắng thường	White flesh Guava	(10°24′51″ N, 105°51′59″ E, morn. 10 m)	PG02
Ruot hong da lang	Ruột hồng da láng	Pink flesh smooth skin Guava	(10°24′49″ N, 105°51′57″ E, morn. 10 m)	PG03
Ruot hong da san	Ruột hồng da sần	Pink flesh rough skin Guava	(10°19′54″ N, 105°54′28″ E, morn. 10 m)	PG04
Taiwan Guava	Ổi Đài Loan (Ổi lê Đài Loan)	Taiwan Guava	(10°20′12″ N, 105°55′1″ E, morn. 10 m)	PG05
Nu hoang	Ổi nữ hoàng	Queen Guava	(10°21′23″ N, 105°53′12″ E, morn. 10 m)	PG06

## Data Availability

Data are available from the corresponding author (H.H.N.) upon reasonable request.

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
