# Peer review of "Chemical Composition, Larvicidal and Molluscicidal Activity of Essential Oils of Six Guava Cultivars Grown in Vietnam"

_plants, 2023, doi:10.3390/plants12152888_

Round 1
Reviewer 1 Report
Find below the necessary revisions to the manuscript:
- The statistical methods used must be inserted in section 4.7
- The conclusions of the manuscript are too bad. In this section, authors must write a single paragraph with: background, main results obtained from all methodologies used, how the results of the manuscript advance the frontier of knowledge on the investigated topic
- In section 4.2 authors must include how the plant material was processed, drying process, cutting plant material among other information.
- In the title of table 1, the authors must report the measurements of the compounds.
- At the beginning of section 2.1, authors must inform the minimum percentage to consider a compound as a major compound.
- The authors presented a wide variety of results. However, the discussion of the results, due to the large volume of data, was not enough. To further improve the quality of the manuscript, it would be important for the authors to place the less relevant results as supplementary material, keeping in the manuscript (results and discussions section) only those really relevant results. Thus, the authors will have a more concise role and with relevant information for the readers of this journal.
Reviewer 2 Report
Review comments on “Chemical composition, larvicidal and molluscicidal activity of essential oils of six guava cultivars grown in Vietnam”
General comments
It is an interesting manuscript to discuss the extracts from the six guava cultivars and their effects on the mosquito control. Two composition-based clusters were detected one of (E)-β-caryophyllene and the other of limonene / (E)-β-caryophyllene, in this study. I think the manuscript should take a mirror revision before the final consideration. Please see the comments below,
1. Please redesign the figure on Page 14;
2. Is there any possible to provide some images of your study, it is better to see some effects from the images.
3. Is there any research about the extract of Guava, I think it is a common species in the world;
4. I am not sure the animal experiment, do we need a certificate for the study?
5. Please update some writing grammar in the M&M part.
Reviewer 3 Report
Manuscript Number: plants-2533023, titled:
Chemical composition, larvicidal and molluscicidal activity of essential oils of six guava cultivars grown in Vietnam
Review 1 – 19 July 2023
Dear Editor of Plants
the argument is interesting but it has to be improved. The introduction has to be better presented. The experimental design has to be detailed the information given is not enough. The manuscript is not always arranged as per the instructions for authors of Plants. Inaccuracies in the text.
I suggest a major revision
To the Authors (in detail):
- the argument is interesting but it has to be improved. The introduction has to be better presented. The experimental design has to be detailed the information given is not enough. The manuscript is not always arranged as per the instructions for authors of Plants. Inaccuracies in the text;
- Introduction section, please, explain: (Chen, 1935) [28];
- Introduction section, Pleas explain that essential oil composition is influenced by many factors, such as: cultivar (genetic aspect) [X1], geographical area of production [X2], extraction system [3], find, read and discuss:
[X1] The peel essential oil composition of bergamot fruit (Citrus bergamia, Risso) of Reggio Calabria (Italy): a review.
Emirates Journal of Food and Agriculture 32 (11) 835-845 (2020)
doi: 10.9755/ejfa.2020.v32.i11.2197
[X2] Determination of the Volatile Composition in Essential Oil of Azadirachta indica A. Juss from different areas of North Indian Plains by Gas Chromatography/Mass Spectrometry (GC/MS), Analytical Chemistry Letters, 11:1, 73-82, DOI: 10.1080/22297928.2021.187719
[X3] Comparison of Essential Oils Obtained from Different Extraction Techniques as an Aid in Identifying Aroma Significant Compounds of Nutmeg (Myristica fragrans).
Natural products Communication 10, 1443-1446 (2015).
- In relation to guava, sometime you use cultivar and sometime variety. Cultivar and variety are not synonyms, please verify and correct in the whole manuscript;
- Tables 4,7, 10 and in the whole manuscript: sometime you have written the chemical compound in capital letter and sometime in small letter, please, standardize;
- Discussion section extend and try to specify the essential oil/s showing the larvicidal effect;
- Sub-section 4.1, detail if plants were wild or cultivated. If cultivated detail: type of fertilizers (type, quantity, period); type of irrigation. In any case detail: type of soil, microclimate, age of plants; days after flowering; time between sample collecting and essential oil extraction; how samples were stored before EO extraction; time between essential oil extraction and analysis and how the essential oil was stored before analysis (type of container and volume of EO and container;
- Sub-section 4.2. please, detail the type of water used and the ratio between sample and water;
- Sub-section 4.6 and in the whole manuscript, when you indicate a temperature, separate the numeric value and the symbol °C;
- Sub-section 4.7 and in the whole manuscript, apply the instructions for authors to incorporate the references;
- Please, write in blue color or evidence differently the corrections you will do.
I suggest a major revision
Regards.
Round 2
Reviewer 1 Report
The authors made the necessary changes. Now the manuscript is ready for publication in this journal.
Reviewer 3 Report
Manuscript Number: plants-2533023, titled:
Chemical composition, larvicidal and molluscicidal activity of essential oils of six guava cultivars grown in Vietnam
Review 2 – 29 July 2023
Dear Editor of Plants
the argument is interesting and the authors have included all my comments. The introduction is well presented. The experimental design is well described. The discussion of data is well argued.
I suggest to publish in the current form
Regards.